# Ecotoxicological and Chemical Approach to Assessing Environmental Effects from Pesticide Use in Organic and Conventional Rice Paddies

Fulvio Onorati [1,*], Andrea Tornambé [1], Andrea Paina [1], Chiara Maggi [1], Giulio Sesta [1], Maria Teresa Berducci [1], Micol Bellucci [2], Enrico Rivella [3] and Susanna D'Antoni [1]

1   Institute for Environmental Protection and Research, 00144 Rome, Italy
2   Research and Science Department, Italian Space Agency, 00133 Rome, Italy
3   Regional Environmental Protection Agency (ARPA), 10135 Torino, Italy
*   Correspondence: fulvio.onorati@isprambiente.it; Tel.: +39-0650074648

**Abstract:** Despite laws and directives for the regulation and restriction of pesticides in farming, the large use of Plant-Protection Products (PPPs) in paddy fields is a relevant worldwide cause of environmental contamination. The aim of this work is to evaluate the environmental impact due to the use of PPPs by using an integrated approach based on chemical analyses and ecotoxicological hazard assessment, supported by statistical tools, in order to overcome the issues related to traditional tabular evaluation. Samples of soil and water of seven conventional and organic paddies located in Northern Italy were examined for two years. The results evidenced a direct relationship between the presence of Oxadiazon in water and bioassay responses as the main cause of the toxicity measured. This phenomenon affected both biological and conventional rice fields, due to the free circulation of water through irrigation canals. Therefore, the implementation of organic districts with water circulation isolated from conventional fields represents a simple and effective countermeasure to safeguard the agricultural practices of organic crops.

**Keywords:** environmental impact assessment; bioassays; ecotoxicological hazard; pesticides; oxadiazon

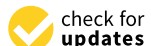



## 1. Introduction

For a long time, it has been known that the large amount of pesticides applied in paddy fields, in addition to the common practice of draining the paddy water in irrigation canals that flows into the freshwater system and eventually into the marine environment, is one of the major causes of pollution worldwide [1].

Plant-Protection Products (PPPs), because of their persistence, toxicity, and bioaccumulative properties, are of particular concern. They might have adverse ecological effects, causing both short-term (acute) and long-term (chronic), lethal or sub-lethal biological damage; in particular, changes in behavior, metabolism, development, alteration in the food chain or habitat of non-target organisms, such as amphibians and bats, and reduction in the populations of natural predators of insect pests [2–17]. In addition, pesticides are known to decrease the biodiversity and function of an ecosystem by promoting the dominance of undesired and invasive species [18].

The most common way to assess the biological effects of the use of PPPs derives from toxicological studies with the aim to define the existing relationship between doses of specific compounds and toxicity response in laboratory experiments. Relatively few field surveys were conducted with the purpose of correlating the in situ measurements of pesticide concentration with the bioassay responses, i.e., *Daphnia magna* [19]. Some studies were carried out in controlled microcosms on a range of representative soil organisms [20], while several studies used the Cornell University Environmental Impact Quotient (EIQ) calculator [21] and the ECOTOX Knowledgebase to determine the exposure risk associated

with individual pesticides relative to their application rates and aquatic concentrations [22]. Moreover, Sánchez-Bayo and Goka [23] evaluated the ability of four community endpoints (species richness, abundance, diversity, and similarity indices) to assess the impacts of two insecticides (Imidacloprid and Etofenprox). The ecotoxicity of a mixture of PPPs and behavior of pesticide transformation products directly in the aquatic environment, such as paddies, are poorly understood.

The European Community established various actions to counter the impact of pesticides on biodiversity through the Directive 2009/128/CE [24], which was implemented in Italy with the Legislative Decree no. 150/2012 [25], compelling a minimization or prohibition of the use of pesticides in the areas designated by the Habitat (92/43/EEC) and Birds (2009/147/EC) Directives [26,27] and in the protected areas referred to in the Water Framework Directive (2000/60/EC) [28]. Europe and United States (USEPA) established Regulation (EC) No. 1107/2009 [29] and the Data Requirements for Conventional Chemicals for pesticide registration, requiring environment fate and ecotoxicity data to be provided [30].

These regulations are referred to as a tabular approach, which is limited to the compliance of the chemical thresholds with respect to the toxicity of pure individual pesticides. Moreover, this approach does not take into account the simultaneous action of the complex mixtures of contaminants commonly present in aquatic habitats that may result in antagonistic and more often synergistic effects on biota.

One of the most advanced pesticide actions is adopted in Japan by the Ministry of Environment [31]. In order to determine the eligibility of the product, acute toxicity tests must be conducted for fish (basically, *Cyprinus carpio*), daphnids (*Daphnia magna*), and algae (*Raphidocelis subcapitata*), and then the minimum value of the 50% effect concentration (EC50 or LC50) is divided by an uncertainty factor that considers the species sensitivity. The Japanese Agricultural Chemicals Regulation Law was revised in 2018, and the method of assessing pesticide registration criteria was also revised [31]. Toxicity tests using aquatic plants, such as *Lemna* sp., in addition to algae will be introduced in the setting of criteria for herbicides. The uncertainty factor applied to the algal EC50 was changed from 1 to 10 by default, which is then reduced depending on the number of algal species tested. However, registration criteria for eligibility of pesticides for the new method were not developed until 2021 [32].

A relatively common way to determine hazardous concentration for the protection of an ecosystem and to reveal ecological risk is the cumulative distribution function called SSD (Species Sensitivity Distribution) [32,33]. The 5th percentile of this distribution (called the 5% Hazardous Concentration, HC5) has been used by USEPA [34], the RIVM Institute (The Netherlands) [35], and the European Commission [36] for deriving threshold concentrations that protect almost all species in a community. Research based on SSD was conducted for Ecological Risk Assessment (ERA) of several paddy insecticides and herbicides applying the Potentially Affected Fraction (PAF) as an index for the magnitude of ecological risk capable of reducing diversity [32,37,38].

The aim of this work is to investigate the relationship between chemical and ecotoxicological Lines of Evidence (LOEs) in the assessment of environmental impact due to the use of PPPs in Italian paddies, supported by a statistical approach, in a more realistic way than the traditional tabular evaluation. The findings will provide scientifically useful indications on the environmental compatibility of the use of pesticides.

The chemical and ecotoxicological analyses were carried out in conjunction with the sampling of species and habitats linked to the agroecosystems, in a larger project aimed at testing the measures envisaged by the National Action Plan for the sustainable use of PPPs for Natura 2000 sites and in protected areas, through the comparison of the values in biological and conventional fields. These measures indicate the biological method as the one most compatible with the conservation of biodiversity.

## 2. Materials and Methods

### 2.1. Area of Study

The study area is located in the Po Valley, between Piedmont and Lombardy (Italy), a vast agricultural territory in which, for irrigation management needs, rice cultivation is mainly monoculture. The cultivation consists of an uninterrupted succession of rice-field chambers interwoven with a large and articulated irrigation network functional for the distribution of water.

Seven rice paddies, belonging to four organic and three conventional farms, located in the Vercelli plain, were investigated from 2018 to 2019 (Figure 1).

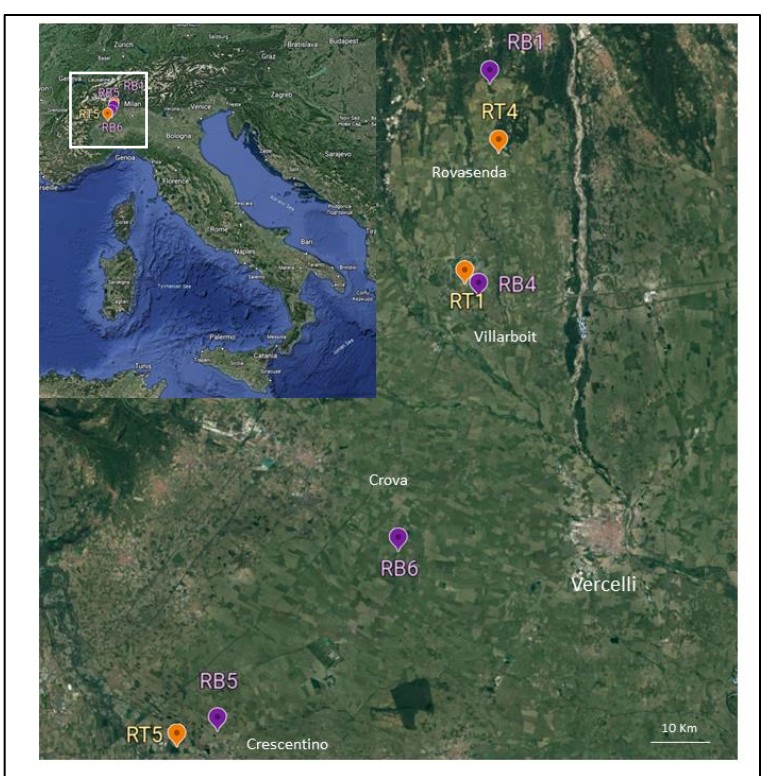

**Figure 1.** Localization of crops involved in the research project.

The zone includes five protected areas and Natura 2000 sites, as detailed in Table 1.

**Table 1.** Identification code, the municipality, and type of agronomic management (RT = conventional; RB = organic) of the seven rice paddies considered in the research.

| Code | Municipality | Protected Areas of the Natura 2000 Network | Agronomic Management | Year |
|------|-------------|---------------------------------------------|----------------------|------|
| RT1 | Villarboit | ZSC/ZPS IT1120014 Druma River marsh | Conventional | 2018–2019 |
| RT4 | Rovasenda | SIC IT1120026 Station of *Isoetes malinverniana* | Conventional | 2018–2019 |
| RT5 | Crescentino | - | Conventional | 2018–2019 |
| RB1 | Rovasenda | ZSC IT1120004 Baraggia of Rovasenda/EUAP0349 Natural Reserve of Baragge | Organic | 2018–2019 |
| RB4 | Villarboit | ZSC/ZPS IT1120014 Druma River marsh | Organic | 2018 |
| RB5 | Crescentino | - | Organic | 2018–2019 |
| RB6 | Crova | ZPS IT1120021 Paddies of Vercelli | Organic | 2019 |

## 2.2. Sampling Strategy

The sampling campaign took place in 2018–2019 and was strongly influenced by the irrigation level and weather conditions. In 2018, soil sampling was performed on the paddy embankment (*em*) and inside the paddy chamber (*ch*), when the growing season was suitable. In 2019, because of long flooding events, samples were collected only in the paddy embankments and the paddy field RB4 was replaced with RB6 (Table 1).

The sampling strategy for water and soil samples was divided into two phases for each year: one sample was collected at the beginning of the growing season, before the phytosanitary treatment ($t_0$), and a second one after the phytosanitary treatments ($t_1$), for a total of four sampling events in two years. For each phase, in presence of water flow, water samples were taken at entry (*in*) and exit (*out*) of the paddy field. In the absence of water flow, two water samples were taken near the channels of entry (*in*) and exit (*out*) from the paddy field. This sampling strategy allowed for verifying the possible contributions of contaminants already present in the irrigation water before entering into the paddy field chamber.

Water samples were kept in decontaminated glass bottles and stored in the dark at −20 °C until further analyses.

Four aliquots of soil from each paddy embankment were collected, one on each side of the chamber; the aliquots were then pooled, homogenized, sieved (2 mm), and stored at −20 °C.

## 2.3. Chemical Characterization

### 2.3.1. Soil Samples

Soil samples were dried and homogenized by grinding with an IKA® mill equipped with a beater blade. Total Carbon (TC) and Total Nitrogen (TN) were determined with a CHNS analyzer Vario Micro Cube Elementar and referred to as dry weight. Inorganic carbon was removed by progressive additions of HCl, then evaporated at ~50 °C. The analyzer performed a controlled combustion (~900 °C), then a catalytic oxidation (Chromium oxide), and finally a reduction by metallic copper. The $CO_2$ and $N_2$ developed were determined by a thermal conductivity detector after their gas chromatographic separation. Quality control was performed by daily Acetanilide analysis and by repeated measurements of two standard soils (Boden Standard AIVA Analysentechnik and Soil standard 2.1 Elementar).

For Ca, K, and Mg analyses, the sample was dried at 35 °C for 48 h and homogenized in an agate mortar. It was then subjected to a microwave acid digestion, using a 1:3 mixture of nitric and hydrochloric acid [39]. Instrumental determination was performed by Inductively Coupled Plasma Atomic Emission Spectroscopy (ICP-OES) [40], at the following wavelengths: 766.492 nm for K; 422.673 nm for Ca; 279.800 nm for Mg, referring concentrations to dry weight. The limit of quantification (LOQ) for these elements was 0.010%. Quality control was performed analyzing certified reference materials (sediment PACS-2 and soil SRM 2709).

The following PPPs were determined in dried and homogenized soil samples: Chlorpyrifos, Penconazole, Metalaxyl, Metrafenone, Pendimenthalin, Metolachlor, Alpha Endosulfan, Beta Endosulfan, Oxadiazon, Boscalid, Deltamethrin, Lambda Cyhalothrin, Oxifluorfen, Tebuconazole, Folpet, Dimethomorph, and Kresoxim methyl.

PPPs were extracted by Pressurized Fluid Extraction, dehydrated, and concentrated under a nitrogen flow, filtered, and analyzed by Gas Chromatography coupled with a triple-quadrupole Mass Spectrometer (GC/MS/MS) in MRM mode. Identification was based on the presence of at least two characteristic transitions for each analyte. Quantification was performed using perdeuterated internal injection standards. The LOQ of the method was 0.1 mg $Kg^{-1}$ for each analyte. The method was developed by recovery tests on spiked samples; the quality control involved the analysis, at each sequence, of blanks, replicates, and recovery tests.

2.3.2. Water Samples

The determination of Ca, K, Mg, and S was performed on whole samples, after acidification with 2% nitric acid, by Atomic Emission Spectroscopy technique with inductively coupled plasma [40] at the same wavelengths as for soil samples. Limits of quantification were 4 mg $L^{-1}$ for Calcium and 1 mg $L^{-1}$ for Mg, K, and S. Copper determination was carried out by AAS technique with a graphite furnace [41] at a wavelength of 327.4 nm, with a quantification limit (LOQ) of 1 μg $L^{-1}$. The quality control was performed using different certified reference materials: NIST 1643f for Ca, K, Mg and Cu, and SRM2709 for S.

The same PPPs of soil were also detected in water samples. A solution of deuterated internal standards was added to 200 mL of aqueous sample, and an extraction with dichloromethane was performed three times by separation funnel shaking. The extract was dehydrated with anhydrous sodium sulfate to the final volume by nitrogen blowing and then analyzed by GC/MS in SIM mode. Identification was performed by comparison of retention indices with those of standards and comparison of the relative abundances of the ions. Quality control was ensured by the analysis, for each batch of samples, of standards, method blanks, and recovery tests on spiked samples. The limit of quantification of the method (LOQ) for Folpet and Deltamethrin was 0.20 μg $L^{-1}$ and for all other substances was 0.10 μg $L^{-1}$.

*2.4. Ecotoxicological Characterization*

Ecotoxicological assays were started simultaneously on all test species, using the same sample aliquot of chemical analyses.

Different batteries of bioassays were set up in order to optimize the ecological representativeness of test species with respect to the environmental characteristics of the crops under study. The main features of the toxicity tests are detailed in Table 2.

**Table 2.** Battery of bioassays used for ecotoxicological assessment of soil and water samples of rice paddies.

| Sampling Point | Species | Common Name | Environmental Matrix | End-Point | Exposition | Method |
|---|---|---|---|---|---|---|
| Embankment and paddy chamber soil | *Lepidium sativum* | Watercress | Soil | Germination and root elongation | 72 h | ISO 18763:2016 |
| | *Sinapis alba* | Mustard | | | | |
| | *Sorghum saccharatum* | Sorghum | | | | |
| | *Aliivibrio fischeri* | Bacterium | Eluate | Biolumin. | 30 min | ISO 11348-3:2019 |
| Water | *Raphidocelis subcapitata* | Green algae | Water | Growth rate | 72 h | ISO 8692:2012 |
| | *Daphnia magna* | Water flea | Water | Immobilization | 24 h | ISO 6341:2013 |
| | *Spirodela polyrhiza* | Duckmeat | Water | Leaves growth | 72 h | ISO 20227:2017 |
| | *Aliivibrio fischeri* | Bacterium | Water | Biolumin. | 30 min | ISO 11348-3:2019 |

All bioassays were performed according to standardized ISO methods in the ISPRA certified laboratories, according to UNI EN ISO 9001:2015.

*2.5. Ecotoxicological Hazard Assessment*

ISPRA and Public University of Ancona (Italy) have already developed several synthetic indices for the ecological risk assessment in marine environments [42,43], transposed in 2016 in the Technical Annex of Ministerial Decree No. 173/2016, which regulates dredging activities in relation to dumping.

The model, by means of a modular structure corresponding to several LOEs, combines, in a weighted way, the chemical, biological, and ecotoxicological characteristics of sediments. Regarding the ecotoxicological LOE, the Hazard Quotient ($HQ_{eco}$) integrates the results of the bioassays based not only on the biological measured effects, but also on

the severity of the endpoint (growth, bioluminescence, survival, embryonic development, etc.), the ecological relevance of the tested environmental matrix (pore water, elutriate, whole sediment, etc.) as well as the type of exposure (chronic or acute) [42]. In particular, weighted criteria to elaborate results from standardized ecotoxicological bioassays are based on specific thresholds and weights assigned to each bioassay depending on the biological endpoint, tested matrix, time of exposure, and the possibility of hormetic responses. The cumulative Hazard Quotient referred to as battery bioassays ($HQ_{Battery}$) is obtained by the summation ($\Sigma$) of the weighted effects ($E_w$), i.e., the variations measured for each test compared to specific thresholds, corrected for the statistical significance of the difference ($w$), the biological importance of the endpoint, and exposure conditions ($w_2$) [42]:

$$HQ_{Battery} = \sum_{K=1}^{N} Effect_w(k) \cdot w_2$$

The $HQ_{Battery}$ is normalized to a scale ranging from 0 to 10, where 1 is the battery threshold when all the bioassays exhibit an effect equal to own toxicity threshold, while 10 indicates that all the assays exhibit 100% of the effect. The $HQ_{Battery}$ is then assigned to one of five classes of hazard, from Absent to Severe [43,44].

The flexible structure of HQ index and weighted criteria can be easily adapted to other applications, simply by varying the test species and weights of the different variables considered environmentally relevant.

In the case of rice fields, the toxicity thresholds and weights assigned to variables considered in the integrated index for estimating ecotoxicological Hazard Quotient ($HQ_{eco}$) are detailed in Table S1 (Supplementary Materials). The weight of the different types of endpoint varies according to the severity of the biological effect, with a maximum value for mortality (2.2) and a minimum value for root elongation (1.1). According to the same principle, acute toxicity has greater weight (1.0) than chronic toxicity (0.8).

### 2.6. Statistical Analyses

In order to compare the ecotoxicological results obtained between organic and conventional paddies, a specific t-test was applied for inhomogeneous variance at a confidence level of 95% ($p = 0.05$). Finally, with the aim to assess possible relationships between the measured ecotoxicological responses and the chemical characteristics detected in the different matrices, a multivariate analysis (PCA) was carried out on the raw data. Statistical elaborations were performed using R (version 4.2.1) and RStudio (version 2022.07.1) with the installed packages *lme4*, *vegan*, *factoextra*, and *ggplot2*.

## 3. Results and Discussion

### 3.1. Chemical Characteristics of Soil

In Table S2 (Supplementary Materials) the raw data referred to as organic content (TOC, TC), macro-elements, and inorganic pesticides (Cu and S) sought for characterization of samples are detailed. The percentages of Total Nitrogen (TN), approximately in a range of 0.1–0.3, reflect a type of soil "well endowed", according to the classification of Giardini [45], while the C/N ratios are generally > 11. The high percentage of organic component leads to nitrogen stabilization, making it less bioavailable. The ratios did not change substantially between organic and conventional paddies, with median values of 11.22 and 10.58, respectively. The amount of organic matter did not show significant differences between the organic and conventional fields ($p = 0.335$) throughout the study period, whereas conventional paddies resulted in significantly higher contents of Cu and S in 2018 ($p = 0.005$ and 0.004, respectively). For both metals, the highest values were always found in the conventionally treated rice fields, especially in RT5. The relatively high values measured could be associated with the large use of copper sulphate, as classic verdigris in the entire area.

Table S3 reports the detail of results for the pesticides for which at least one sample showed quantifiable concentrations (>LOQ). In particular, the PPP concentrations were <LOQ in all samples from organic crops, while a few cases of barely quantifiable concentrations of Oxadiazon, Oxyfluorfen, and Pendimethalin were found in samples from conventional paddies. The low number of quantifiable results does not allow for evaluating differences between the various types of samples from rice fields (chamber, embankment, soil inlet, or outlet) or even just between before ($t_0$) and after treatment ($t_1$).

Despite the many active ingredients sought, most measures were below the LOQ. The finding demonstrates that, despite frequent treatments with PPPs over the years, the irrigation cycles in rice fields favor soil leaching, which, therefore, retains little residue of contaminants in the chamber, promoting their transfer to the water.

### 3.2. Chemical Characteristics of Water

Table 3 shows the concentrations of macro elements (Ca, K, Mg) and inorganic and organic pesticides measured in the water samples for which at least one value was quantified, except for sample $RT4\_in\_t_0$ because its amount was not sufficient to carry out the analyses.

**Table 3.** Chemical characterization of water samples from rice paddies (RB = organic cultures; RT = conventional cultures). In 2019 RB4 was substitute with RB6 (in = paddy field chamber entrance; out = paddy field chamber exit).

| | Sample | | Ca (mg L$^{-1}$) | K (mg L$^{-1}$) | Mg (mg L$^{-1}$) | Cu (µg L$^{-1}$) | S (mg L$^{-1}$) | Λ-Cyalothrin (µg L$^{-1}$) | Metolachlor (µg L$^{-1}$) | Oxadiazon (µg L$^{-1}$) | Oxyfluorfen (µg L$^{-1}$) |
|---|---|---|---|---|---|---|---|---|---|---|---|
| | | | | | | **2018** | | | | | |
| RB1 | in | $t_0$ | 4.81 | 1.07 | 1.12 | 3.60 | 1.79 | <0.1 | <0.1 | 0.15 | <0.1 |
| | | $t_1$ | 8.86 | 2.78 | 2.02 | 2.85 | 2.31 | <0.1 | <0.1 | 0.14 | <0.1 |
| | out | $t_0$ | 9.44 | 19.79 | 3.08 | 8.07 | 2.44 | <0.1 | <0.1 | 0.13 | <0.1 |
| | | $t_1$ | 12.76 | <1.00 | 3.57 | <1.00 | <1.00 | <0.1 | <0.1 | 0.10 | <0.1 |
| RB4 | in | $t_0$ | 9.22 | 5.57 | 2.25 | 1.85 | 2.7 | <0.1 | <0.1 | 1.68 | <0.1 |
| | | $t_1$ | 11.00 | 4.00 | 3.1 | 1.94 | 1.17 | <0.1 | <0.1 | 0.67 | <0.1 |
| | out | $t_0$ | 13.85 | 3.81 | 3.31 | 6.86 | 3.13 | <0.1 | <0.1 | 0.21 | <0.1 |
| | | $t_1$ | 9.89 | 1.90 | 2.81 | 5.67 | <1.00 | <0.1 | <0.1 | 0.52 | <0.1 |
| RB5 | in | $t_0$ | 27.39 | 6.92 | 8.00 | 4.92 | 10.08 | <0.1 | <0.1 | 1.72 | <0.1 |
| | | $t_1$ | 34.05 | <1.00 | 3.57 | 2.10 | 5.06 | <0.1 | <0.1 | 1.09 | <0.1 |
| | out | $t_0$ | 36.59 | 8.24 | 8.22 | 2.73 | 7.01 | <0.1 | <0.1 | 0.52 | <0.1 |
| | | $t_1$ | 36.59 | 1.53 | 9.99 | 3.03 | 19.58 | <0.1 | <0.1 | 0.23 | <0.1 |
| RT1 | in | $t_0$ | 6.29 | 2.09 | 1.34 | 2.67 | 2.27 | <0.1 | <0.1 | 1.75 | 0.23 |
| | | $t_1$ | 10.9 | 2.79 | 2.56 | 5.58 | 2.44 | <0.1 | <0.1 | 0.61 | <0.1 |
| | out | $t_0$ | 5.63 | 2.40 | 1.09 | 5.56 | 3.12 | <0.1 | <0.1 | 1.24 | 0.12 |
| | | $t_1$ | 9.52 | 33.93 | 3.05 | 6.69 | 1.50 | <0.1 | <0.1 | 0.35 | <0.1 |
| RT4 | in | $t_0$ | n.d. | n.d. | n.d. | n.d. | n.d. | <0.1 | <0.1 | 0.13 | <0.1 |
| | | $t_1$ | 5.60 | <1.00 | 1.22 | 6.73 | 1.63 | <0.1 | <0.1 | 0.17 | <0.1 |
| | out | $t_0$ | 4.27 | 1.96 | <1.00 | 3.82 | 2.43 | <0.1 | <0.1 | 0.15 | <0.1 |
| | | $t_1$ | 6.14 | 1.31 | 1.85 | 7.22 | <1.00 | <0.1 | <0.1 | 0.42 | <0.1 |
| RT5 | in | $t_0$ | 42.66 | 3.58 | 8.72 | 2.12 | 10.42 | <0.1 | <0.1 | 0.82 | <0.1 |
| | | $t_1$ | 30.73 | 3.85 | 15.01 | <1.00 | 9.25 | <0.1 | 0.13 | 0.22 | <0.1 |
| | out | $t_0$ | 25.13 | 16.9 | 6.68 | 4.56 | 9.31 | <0.1 | <0.1 | 0.26 | <0.1 |
| | | $t_1$ | 30.16 | 3.13 | 6.89 | 2.59 | 6.92 | <0.1 | 0.47 | 0.35 | <0.1 |
| | | | | | | | **2019** | | | | | |
| RB1 | in | $t_0$ | 8.02 | 6.4 | 1.74 | 1.03 | 1.88 | <0.1 | <0.1 | 1.56 | <0.1 |
| | | $t_1$ | 9.58 | 2.03 | 1.93 | 1.16 | 1.61 | <0.1 | <0.1 | 0.11 | <0.1 |
| | out | $t_0$ | 12.19 | 13.35 | 3.19 | 2.75 | 2.81 | <0.1 | <0.1 | <0.1 | <0.1 |
| | | $t_1$ | 8.95 | 2.00 | 1.80 | 1.29 | 1.31 | <0.1 | <0.1 | 0.17 | <0.1 |

**Table 3.** *Cont.*

| | Sample | | Ca (mg L$^{-1}$) | K (mg L$^{-1}$) | Mg (mg L$^{-1}$) | Cu (µg L$^{-1}$) | S (mg L$^{-1}$) | Λ-Cyalothrin (µg L$^{-1}$) | Metolachlor (µg L$^{-1}$) | Oxadiazon (µg L$^{-1}$) | Oxyfluorfen (µg L$^{-1}$) |
|---|---|---|---|---|---|---|---|---|---|---|---|
| RB5 | in | $t_0$ | 39.14 | 4.57 | 9.54 | 1.92 | 16.36 | <0.1 | <0.1 | 0.26 | <0.1 |
| | | $t_1$ | 44.25 | 5.99 | 11.3 | 2.34 | 10.08 | <0.1 | <0.1 | 0.18 | <0.1 |
| | out | $t_0$ | 42.17 | 16.74 | 8.11 | 2.23 | 7.41 | <0.1 | <0.1 | <0.1 | <0.1 |
| | | $t_1$ | 31.53 | 6.28 | 9.95 | <1.00 | 4.32 | <0.1 | <0.1 | 0.23 | <0.1 |
| RB6 | in | $t_0$ | 15.26 | 1.71 | 3.45 | 1.06 | 5.78 | 0.17 | <0.1 | 0.20 | <0.1 |
| | | $t_1$ | 17.70 | 1.08 | 4.73 | <1.00 | 4.85 | <0.1 | <0.1 | <0.1 | <0.1 |
| | out | $t_0$ | 13.88 | 1.98 | 3.11 | <1.00 | 6.61 | <0.1 | <0.1 | 0.43 | <0.1 |
| | | $t_1$ | 17.46 | <1.00 | 3.98 | <1.00 | 3.81 | <0.1 | <0.1 | 0.46 | <0.1 |
| RT1 | in | $t_0$ | 6.05 | 1.89 | 1.35 | <1.00 | 2.34 | <0.1 | <0.1 | 0.13 | <0.1 |
| | | $t_1$ | 6.58 | 2.65 | 1.37 | 1.05 | <1.00 | <0.1 | <0.1 | 0.37 | <0.1 |
| | out | $t_0$ | 5.43 | 3.32 | 1.11 | 6.86 | 3.2 | <0.1 | <0.1 | <0.1 | <0.1 |
| | | $t_1$ | 12.15 | <1.00 | 2.34 | 1.14 | <1.00 | <0.1 | <0.1 | 0.40 | <0.1 |
| RT4 | in | $t_0$ | 6.47 | <1.00 | 1.04 | <1.00 | 1.90 | <0.1 | <0.1 | 0.17 | <0.1 |
| | | $t_1$ | 6.37 | 1.40 | 1.41 | 1.23 | 2.06 | <0.1 | <0.1 | <0.1 | <0.1 |
| | out | $t_0$ | 5.11 | 1.57 | 1.06 | 2.63 | 3.01 | <0.1 | <0.1 | <0.1 | <0.1 |
| | | $t_1$ | 6.11 | 1.11 | 1.11 | 1.44 | 2.11 | <0.1 | <0.1 | 0.14 | <0.1 |
| RT5 | in | $t_0$ | 29.79 | 3.47 | 8.47 | 1.30 | 15.28 | <0.1 | <0.1 | 0.29 | <0.1 |
| | | $t_1$ | 16.98 | 2.99 | 4.67 | <1.00 | 6.17 | <0.1 | <0.1 | 0.29 | <0.1 |
| | out | $t_0$ | 27.45 | 4.55 | 4.73 | 2.24 | 12.97 | <0.1 | <0.1 | 47.6 | <0.1 |
| | | $t_1$ | 24.07 | 5.61 | 7.69 | 2.47 | 7.64 | <0.1 | <0.1 | 0.71 | <0.1 |

The concentrations of the macro elements indicate that Ca is the dominant element (4.27–44.25 mg L$^{-1}$), followed by Mg (up to 15.01 mg L$^{-1}$) and K (up to 33.93 mg L$^{-1}$) in similar quantities. No significant statistical differences were found between organic and conventional fields about the content of both macro elements and inorganic pesticides. However, a general trend is noticeable towards a lower Ca, Mg, and K content in organic rice fields, especially in 2019 ($p = 0.076$ for Ca; $p = 0.097$ for K; $p = 0.099$ for Mg).

No differences related to either the sampling point (in or out) or the two sampling campaigns following treatment were noticed. The concentrations of Cu and S measured in water are in very low ranges, especially for copper, as are the values found in soil samples from the same fields. This is likely due to the limitations on the use of copper sulphate in rice fields from 2012, according to Regulation 1107/2009/EC [29].

Oxadiazon was detected in almost all samples analyzed in concentrations up to 47.6 µg L$^{-1}$ (RT5_out_$t_0$), showing widespread contamination, although the organic paddies showed concentrations generally lower (0.45 ± 0.52 µg L$^{-1}$) than conventional ones (2.36 ± 9.44 µg L$^{-1}$), but still present, even before the first treatment of the culture cycle ($t_0$). The overall variability among the rice fields is such that no statistically significant difference was identified ($p = 0.343$), considering both the individual years of investigation and the overall comparison among organic and conventional crops.

Other pesticides, such as Metolachlor and Oxyfluorfen, were detected in very low concentrations in sporadic cases (Table 3), whereas the other chemicals analyzed were all below the LOQ of the specific method, except for lambda-cyhalothrin in RB6_in_$t_0$ (0.17 µg L$^{-1}$).

*3.3. Ecotoxicological Effects of Soil*

The soil samples of conventional paddies showed significant toxicity for root elongation (Table 4) and for the bioluminescence inhibition assays on eluate (Table 5), in particular for RT5 and RT4 rice fields, making the difference between organic and conventional crops statistically significant for *L. sativum* ($p = 0.047$) and *S. saccharatum* ($p = 0.001$).

**Table 4.** Results of phytotoxicity tests with *Lepidium sativum* (Ls), *Sinapis alba* (Sa), *Sorghum saccharatum* (Ss) on samples of soil collected during campaigns $t_0$ and $t_1$ (in bold are shown the significative effects above the toxicity threshold). Negative sign indicates biostimulation of root growth in comparison to control (Ch = rise chamber; em = embankment).

| 2018 | | | Ls (%) | ±sd | Sa (%) | ±sd | Ss % | ±sd | 2018 | | | Ls (%) | ±sd | Sa (%) | ±sd | Ss (%) | ±sd |
|---|---|---|---|---|---|---|---|---|---|---|---|---|---|---|---|---|---|
| **Conventional** | | | | | | | | | **Organic** | | | | | | | | |
| RT1 | ch | $t_0$ | 17.81 | 7.61 | 1.84 | 2.75 | −48.62 | 7.88 | $t_0$ | em | RB4 | **30.48** | 12.24 | 21.80 | 9.86 | 19.47 | 7.04 |
| RT1 | ch | $t_1$ | **38.20** | 4.88 | 24.53 | 7.83 | 4.66 | 12.09 | $t_1$ | em | RB4 | 26.94 | 17.37 | 15.51 | 14.62 | 21.04 | 15.51 |
| RT4 | em | $t_0$ | 25.84 | 13.36 | **36.27** | 14.63 | **40.81** | 3.88 | $t_0$ | em | RB1 | 18.11 | 14.57 | −6.79 | 10.25 | −12.43 | 22.14 |
| RT4 | em | $t_1$ | n.d. | n.d. | n.d. | n.d. | n.d. | n.d. | $t_1$ | em | RB1 | 23.67 | 2.34 | −1.50 | 11.38 | −11.97 | 13.09 |
| RT4 | ch | $t_0$ | **30.91** | 15.18 | 29.94 | 10.42 | **35.84** | 15.39 | $t_0$ | ch | RB1 | −4.05 | 8.69 | 3.13 | 15.74 | −20.45 | 24.46 |
| RT5 | em | $t_1$ | **94.34** | 0.58 | **82.90** | 3.25 | **32.62** | 12.63 | $t_0$ | em | RB5 | 24.41 | 10.99 | 7.69 | 8.27 | 23.60 | 14.84 |
| RT5 | ch | $t_0$ | 25.15 | 6.45 | 8.28 | 6.38 | 15.96 | 1.02 | | | | | | | | | |
| RT5 | ch | $t_1$ | **59.71** | 9.35 | **53.18** | 6.48 | 24.62 | 7.46 | $t_1$ | em | RB5 | **57.66** | 14.97 | 23.59 | 6.33 | 19.57 | 23.58 |
| **2019** | | | **Ls** | | **Sa** | | **Ss** | | **2019** | | | **Ls** | | **Sa** | | **Ss** | |
| RT1 | em | $t_0$ | 24.18 | 5.92 | **41.66** | 5.33 | 17.52 | 12.06 | $t_0$ | em | RB6 | 20.26 | 3.96 | 20.71 | 16.14 | 14.02 | 2.83 |
| RT1 | em | $t_1$ | 49.06 | 3.37 | **59.85** | 6.02 | 24.37 | 5.53 | $t_1$ | em | RB6 | 6.07 | 8.59 | 9.04 | 11.63 | −10.85 | 5.62 |
| RT4 | em | $t_0$ | 22.40 | 5.52 | 21.14 | 2.96 | 1.60 | 14.33 | $t_0$ | em | RB1 | 28.56 | 2.60 | −8.25 | 10.64 | 5.54 | 3.60 |
| RT4 | em | $t_1$ | 18.19 | 2.79 | **34.99** | 10.24 | 8.75 | 8.11 | $t_1$ | em | RB1 | **32.26** | 2.70 | 6.97 | 20.02 | 5.90 | 5.22 |
| RT5 | em | $t_0$ | **30.55** | 10.42 | 23.79 | 0.13 | 19.44 | 2.39 | $t_0$ | em | RB5 | 21.11 | 16.30 | 10.62 | 13.76 | 14.08 | 11.87 |
| RT5 | em | $t_1$ | **72.78** | 4.83 | **73.08** | 2.18 | **58.79** | 5.15 | $t_1$ | em | RB5 | 15.85 | 9.83 | 11.19 | 10.04 | 13.47 | 8.86 |

**Table 5.** Results of bioassays with *Aliivibrio fischeri* on samples of soil eluate collected during campaigns $t_0$ and $t_1$ (in bold are shown the effects above the toxicity threshold). Negative sign indicates biostimulation in comparison to control (Ch = rise chamber; em = embankment).

| | | | 2018 | | | | | | | |
|---|---|---|---|---|---|---|---|---|---|---|
| **Conventional crops** | | | % | ±sd | **Organic crops** | | | (%) | ±sd |
| RT1 | ch | $t_0$ | **−15.60** | 7.92 | RB4 | em | $t_0$ | **21.50** | 9.50 |
| RT1 | ch | $t_1$ | **31.67** | 20.65 | RB4 | em | $t_1$ | −11.32 | 18.49 |
| RT4 | em | $t_0$ | −3.51 | 25.78 | RB1 | em | $t_0$ | −0.24 | 15.68 |
| RT4 | ch | $t_0$ | −11.04 | 21.68 | RB1 | em | $t_1$ | −28.70 | 36.55 |
| | | | | | RB1 | ch | $t_0$ | **25.32** | 14.53 |
| RT5 | em | $t_0$ | **−18.65** | 19.62 | RB5 | ch | $t_0$ | 10.88 | 7.90 |
| RT5 | ch | $t_0$ | **25.26** | 14.42 | RB5 | em | $t_0$ | −24.77 | 39.92 |
| RT5 | ch | $t_1$ | 1.16 | 21.12 | RB5 | em | $t_1$ | 2.69 | 4.60 |
| | | | 2019 | | | | | | |
| RT1 | em | $t_0$ | 10.00 | 18.21 | RB6 | em | $t_0$ | **−90.59** | 107.65 |
| RT1 | em | $t_1$ | −11.28 | 32.56 | RB6 | em | $t_1$ | −11.84 | 13.97 |
| RT4 | em | $t_0$ | **30.57** | 33.73 | RB1 | em | $t_0$ | **−26.33** | 58.08 |
| RT4 | em | $t_1$ | 10.56 | 14.81 | RB1 | em | $t_1$ | −0.33 | 11.21 |
| RT5 | em | $t_0$ | **−24.17** | 33.99 | RB5 | em | $t_0$ | −3.23 | 8.73 |
| RT5 | em | $t_1$ | **−55.48** | 78.87 | RB5 | em | $t_1$ | **−18.89** | 11.82 |

These results agree with the chemical data measured in conventional paddies regarding traces of Oxadiazon (RT1, RT4 and RT5) and Oxyfluorfen (RT1 and RT5). These PPPs are two herbicides, and the Oxadiazon could be used in Italy until 30 June 2020, whereas Oxyfluorfen is authorized until 2024 [46].

*3.4. Ecotoxicological Effects of Water*

Table 6 reports the results of bioassays expressed as bioluminescence inhibition for *A. fischeri*, average immobilization for *D. magna*, growth rate inhibition for *R. subcapitata*, and leaf growth inhibition for *S. polyrhiza* (only for 2018), respectively.

**Table 6.** Bioassay results expressed as percent effect (%) with standard deviation (sd) on paddy water samples collected during campaigns $t_0$ and $t_1$ before and after PPPs treatment; in and out labels refer to samples taken from streams entering and exiting the paddy water chamber; boldface indicates inhibition effects above the toxicity thresholds; negative sign indicates biostimulation in comparison to the control.

| *Aliivibrio fischeri* | | | | | | | | | | | | |
|---|---|---|---|---|---|---|---|---|---|---|---|---|
| **Conventional** | | | **2018** | | **2019** | | **Organic** | | | **2018** | | **2019** | |
| | | | **(%)** | **ds** | **(%)** | **ds** | | | | **(%)** | **ds** | **(%)** | **ds** |
| RT1 | in | $t_0$ | −11.1 | 8.4 | −4.00 | 10.00 | $t_0$ | in | RB4 (2018) RB6 (2019) | −6.5 | 6.8 | **96.13** | 10.24 |
| | out | | −8.9 | 7.4 | **95.99** | 10.60 | | out | | 14.4 | 13.3 | −7.36 | 7.97 |
| | in | $t_1$ | −5.4 | 12.8 | −4.28 | 14.08 | $t_1$ | in | | −3.1 | 8.5 | −19.61 | 13.75 |
| | out | | −5.9 | 5.4 | −12.05 | 9.74 | | out | | −7.4 | 5.5 | −26.92 | 17.46 |
| RT4 | in | $t_0$ | −11.2 | 20.8 | 3.11 | 17.97 | $t_0$ | in | RB1 | −8.2 | 11.9 | **94.04** | 11.48 |
| | out | | −1.5 | 8.3 | **94.95** | 11.51 | | out | | **21.2** | 26.9 | **90.22** | 17.59 |
| | in | $t_1$ | −7.1 | 10.1 | −5.35 | 6.57 | $t_1$ | in | | −10.0 | 11.0 | **81.88** | 18.99 |
| | out | | −0.8 | 10.1 | **97.16** | 7.52 | | out | | **60.1** | 24.7 | **46.99** | 10.76 |
| RT5 | in | $t_0$ | 1.3 | 11.4 | **69.53** | 13.28 | $t_0$ | in | RB5 | 4.7 | 9.1 | **32.38** | 10.02 |
| | out | | 7.7 | 4.3 | **77.95** | 19.08 | | out | | 4.9 | 14.7 | **39.99** | 21.93 |
| | in | $t_1$ | −6.4 | 10.4 | **36.77** | 10.96 | $t_1$ | in | | −17.1 | 17.7 | **38.67** | 7.59 |
| | out | | −15.0 | 14.9 | **42.00** | 5.00 | | out | | 7.9 | 10.1 | **37.85** | 9.30 |
| *Daphnia magna* | | | | | | | | | | | | |
| **Conventional** | | | **2018** | | **2019** | | **Organic** | | | **2018** | | **2019** | |
| | | | **(%)** | **ds** | **(%)** | **ds** | | | | **(%)** | **ds** | **(%)** | **ds** |
| RT1 | in | $t_0$ | **37.0** | 51.8 | 0.0 | 0.0 | $t_0$ | in | RB4 (2018) RB6 (2019) | **25.0** | 27.8 | 0.0 | 0.0 |
| | out | | **97.0** | 7.1 | 0.0 | 0.0 | | out | | 7.5 | 10.4 | 0.0 | 0.0 |
| | in | $t_1$ | 0.0 | 0.0 | 0.0 | 0.0 | $t_1$ | in | | 0.0 | 0.0 | 0.0 | 0.0 |
| | out | | 7.5 | 10.4 | 0.0 | 0.0 | | out | | 2.5 | 7.1 | 0.0 | 0.0 |
| RT4 | in | $t_0$ | **62.5** | 51.8 | 5.0 | 0.0 | $t_0$ | in | RB1 | 0.0 | 0.0 | 5.0 | 10.0 |
| | out | | 15.0 | 23.3 | 0.0 | 0.0 | | out | | **97.5** | 7.1 | **55.0** | 19.1 |
| | in | $t_1$ | 0.0 | 0.0 | 0.0 | 0.0 | $t_1$ | in | | 2.5 | 7.1 | **45.0** | 19.1 |
| | out | | **50.0** | 53.5 | 0.0 | 0.0 | | out | | 15.0 | 9.3 | **75.0** | 19.1 |
| RT5 | in | $t_0$ | **25.0** | 20.7 | 15.0 | 10.0 | $t_0$ | in | RB5 | 0.0 | 0.0 | 15.0 | 19.1 |
| | out | | 5.0 | 9.3 | **45.0** | 34.2 | | out | | 0.0 | 0.0 | **40.0** | 23.1 |
| | in | $t_1$ | 0.0 | 0.0 | **25.0** | 19.1 | $t_1$ | in | | 0.0 | 0.0 | 20.0 | 16.3 |
| | out | | 0.0 | 0.0 | **25.0** | 10.0 | | out | | 2.5 | 7.1 | 0.0 | 0.0 |

**Table 6.** *Cont.*

| | | | *Raphidocelis subcapitata* | | | | | | | | | |
|---|---|---|---|---|---|---|---|---|---|---|---|---|
| **Conventional** | | | **2018** | | **2019** | | **Organic** | | | **2018** | | **2019** | |
| | | | (%) | ds | (%) | ds | | | | (%) | ds | (%) | ds |
| RT1 | in | $t_0$ | **92.0** | 2.1 | **96.8** | 1.7 | $t_0$ | in | RB4 (2018) RB6 (2019) | **100.0** | 11.0 | **95.0** | 1.2 |
| | out | | **91.1** | 2.9 | **97.2** | 1.9 | | out | | **100.0** | 4.3 | **95.3** | 3.1 |
| | in | $t_1$ | **87.0** | 0.6 | **98.3** | 1.7 | $t_1$ | in | | **89.0** | 0.6 | **99.4** | 2.8 |
| | out | | **100.0** | 3.8 | **97.6** | 4.4 | | out | | **86.0** | 2.1 | **99.4** | 2.0 |
| RT4 | in | $t_0$ | **100.0** | 7.6 | **89.1** | 4.5 | $t_0$ | in | RB1 | **100.0** | 10.3 | **88.5** | 3.9 |
| | out | | **93.0** | 2.3 | **97.8** | 3.5 | | out | | **84.5** | 5.1 | **70.3** | 7.3 |
| | in | $t_1$ | **87.0** | 3.3 | **94.5** | 1.7 | $t_1$ | in | | **100.0** | 7.2 | **100.0** | 4.6 |
| | out | | **87.3** | 2.4 | **80.5** | 3.9 | | out | | **90.0** | 11.9 | **77.1** | 8.3 |
| RT5 | in | $t_0$ | **97.5** | 2.9 | **55.1** | 8.3 | $t_0$ | in | RB5 | **84.1** | 2.2 | **100.0** | 3.7 |
| | out | | **96.3** | 3.8 | **36.6** | 4.5 | | out | | **93.0** | 2.3 | **78.4** | 3.7 |
| | in | $t_1$ | **95.5** | 1.7 | **99.0** | 5.4 | $t_1$ | in | | **94.3** | 1.0 | **51.4** | 5.9 |
| | out | | **94.8** | 1.4 | **75.3** | 5.7 | | out | | **98.7** | 2.2 | **76.0** | 2.6 |

| | | | *Spirodela polyrhiza* | | | | | |
|---|---|---|---|---|---|---|---|---|
| **Conventional** | | | **2018** | | **Organic** | | | **2018** | |
| | | | (%) | sd (%) | | | | (%) | sd (%) |
| RT1 | in | $t_0$ | **−37.9** | 33.8 | $t_0$ | in | RB4/RB6 | −16.3 | 47.2 |
| | out | | −0.6 | 23.8 | | out | | −5.1 | 35.6 |
| | in | $t_1$ | −10.4 | 34.4 | $t_1$ | in | | −19.5 | 26.8 |
| | out | | −8.0 | 34.5 | | out | | −11.8 | 37.2 |
| RT4 | in | $t_0$ | −15.6 | 28.0 | $t_0$ | in | RB1 | −44.6 | 51.0 |
| | out | | −2.6 | 30.5 | | out | | **−35.9** | 32.9 |
| | in | $t_1$ | 12.7 | 32.7 | $t_1$ | in | | −19.5 | 45.4 |
| | out | | **−34.4** | 27.2 | | out | | **−30.2** | 42.6 |
| RT5 | in | $t_0$ | **25.6** | 20.4 | $t_0$ | in | RB5 | **81.9** | 6.8 |
| | out | | 10.0 | 27.7 | | out | | **51.5** | 10.2 |
| | in | $t_1$ | −0.2 | 26.2 | $t_1$ | in | | −1.7 | 27.8 |
| | out | | **23.7** | 23.0 | | out | | −3.3 | 31.9 |

In 2018, with regard to *A. fischeri*, toxicity effects were found only in RB1, with particular reference to water leaving the organic rice chamber; regarding the 2019 campaign, most of the samples showed significant inhibition of bioluminescence, with no statistical differences between conventional and organic crops ($p > 0.05$).

Despite being the least sensitive among the organisms of the selected battery, *D. magna* showed some important toxicity effects on both conventional and organic paddies for both monitoring campaigns. It should be noted that the toxicity data on crustaceans are particularly relevant, since, from an ecotoxicological point of view, they are organisms characterized by relatively low sensitivity but, at the same time, high ecological value for the environment in question, as they live in the aquatic environments of rice paddies that adopt organic cultivation methods [47].

The green alga *R. subcapitata* was the most sensitive species in the battery of bioassays used: for all monitoring campaigns, water samples showed high toxicity, but without particular differences between inlet and outlet water samples, between conventional and

organic rice fields. These ecotoxicological effects could be associated with the presence of at least three herbicides: Metolachlor, Oxadiazon, and Oxifluorfen. In particular, Oxadiazon is found in all organic and treated fields (Table 3), although these substances have not been officially used in organic rice fields. Oxadiazon is known to have important toxic effects on *R. subcapitata* [48]. The high toxicity could, therefore, be linked to the simultaneous presence of these herbicides in the water, considering other possible synergistic effects with other contaminants not researched in this study, but found in the past in paddy field waters in Piedmont (https://www.snpambiente.it/2017/07/11/monitoraggio-dei-fitofarmaci-delle-acque-piemontesi-nelle-risaie/, accessed on 30 November 2022).

Except for sample taken during $t_0$ in paddy RB5, for *S. polyrrhiza*, no inhibition values were revealed, but a general tendency towards biostimulation, probably due to the presence of nutrients in the water was observed.

### 3.5. Ecotoxicological Hazard Index and Statistical Analysis

The results of bioassay applied to embankment, chamber soil, and water samples were elaborated by the ecotoxicological Hazard Index, producing the hazard levels shown in Table 7.

**Table 7.** Ecotoxicological hazard ($HQ_{eco}$) applied to paddy field soil samples. The index is referred to as a bioassay battery with 7 test species. Colors refer to the level of hazard as follows: green = absent; orange = moderate; red = major; black = severe.

| | 2018 | | | | | | | 2019 | | | | | |
|---|---|---|---|---|---|---|---|---|---|---|---|---|---|
| ID | Soil Sampling Point | Campaign | HQeco | Water Sampling Point | Campaign | HQeco | ID | Soil Sampling Point | Campaign | HQeco | Water Sampling Point | Campaign | HQeco |
| RB1 | em | $t_0$ | 0.16 | in | $t_0$ | 2.37 | RB1 | em | $t_0$ | 0.10 | in | $t_0$ | 5.13 |
| | ch | $t_0$ | 0.14 | out | $t_0$ | 8.37 | | | | | out | $t_0$ | 7.32 |
| | em | $t_1$ | 0.09 | in | $t_1$ | 2.41 | | | $t_1$ | 0.17 | in | $t_1$ | 6.29 |
| | | | | out | $t_1$ | 4.64 | | | | | out | $t_1$ | 7.42 |
| RB4 | em | $t_0$ | 0.59 | in | $t_0$ | 3.6 | RB5 | em | $t_0$ | 0.10 | in | $t_0$ | 3.63 |
| | | | | out | $t_0$ | 3.22 | | | | | out | $t_0$ | 5.96 |
| | | $t_1$ | 0.13 | in | $t_1$ | 2.37 | | | $t_1$ | 0.05 | in | $t_1$ | 3.94 |
| | | | | out | $t_1$ | 2.41 | | | | | out | $t_1$ | 3.69 |
| RB5 | em | $t_0$ | 0.07 | in | $t_0$ | 2.42 | RB6 | em | $t_0$ | 0.38 | in | $t_0$ | 5.06 |
| | | | | out | $t_0$ | 3.01 | | | | | out | $t_0$ | 2.37 |
| | | $t_1$ | 0.30 | in | $t_1$ | 2.37 | | | $t_1$ | 0.01 | in | $t_1$ | 2.37 |
| | | | | out | $t_1$ | 2.59 | | | | | out | $t_1$ | 2.37 |
| RT1 | ch | $t_0$ | 0.01 | in | $t_0$ | 3.14 | RT1 | em | $t_0$ | 0.35 | in | $t_0$ | 2.37 |
| | | | | out | $t_0$ | 7.16 | | | | | out | $t_0$ | 5.06 |
| | | $t_1$ | 0.68 | in | $t_1$ | 2.37 | | | $t_1$ | 0.45 | in | $t_1$ | 2.37 |
| | | | | out | $t_1$ | 2.55 | | | | | out | $t_1$ | 2.37 |
| RT4 | em | $t_0$ | 0.29 | in | $t_0$ | 5.48 | RT4 | em | $t_0$ | 0.66 | in | $t_0$ | 5.06 |
| | | | | out | $t_0$ | 2.66 | | | | | in | | 2.46 |
| | ch | $t_1$ | 0.25 | in | $t_1$ | 2.37 | | | $t_1$ | 0.21 | in | $t_1$ | 2.37 |
| | | | | out | $t_1$ | 4.84 | | | | | out | | 5.06 |
| RT5 | em | $t_0$ | 0.71 | in | $t_0$ | 3.61 | RT5 | em | $t_0$ | 0.26 | in | $t_0$ | 4.62 |
| | | $t_0$ | 0.57 | out | $t_0$ | 2.69 | | | | | out | $t_0$ | 3.49 |
| | ch | $t_1$ | 0.50 | in | $t_1$ | 2.37 | | | $t_1$ | 0.98 | in | $t_1$ | 4.15 |
| | | | | out | $t_1$ | 2.37 | | | | | out | $t_1$ | 4.95 |

With regard to the soil samples, no ecotoxicological hazard was detected (HQ < 1), due to a general absence of toxicity, according with the low presence of pesticides (Table S3). Therefore, neither differences between organic fields and their corresponding conventional fields, nor differences between the two survey campaigns, can be highlighted. This is in perfect agreement with the general absence of pesticide residues in soil and sediment samples.

The situation is quite different for water samples. In general, important toxic effects were found on several organisms and for several endpoints, such as to determine an ecotoxicological hazard in all samples (HQ > 1) without exception, sometimes even "major" or "severe" (Table 7). The ecotoxicological hazard affects both conventional and organic paddy waters, with the highest hazard measured in the RB1 field, especially in 2019.

This ecotoxicological picture is also in agreement with the findings of the chemical analysis, with particular reference to the ubiquitous presence of Oxadiazon and Oxyfluorfen.

Figure 2 shows the plot of a Principal Component Analysis (PCA) applied to raw data that explains 67.7% of total variance. Almost all samples appear to be distributed along a gradient oriented with respect to the first component (PCA1), whose main variable contributing to this distribution is the parameter Ca (with a higher factor loading of $-0.924$) and secondly Mg and S in the same way (factor loading of $-0.241$ and $-0.251$, respectively), in accordance with the significantly higher content of these macro-elements in organic crops. With respect to the second component (PCA2), the sample RT5_t0_out_2019 is clearly distinguished in the factorial space due to the highest Oxadiazon concentration, associated to the highest factor loading ($-0.940$).

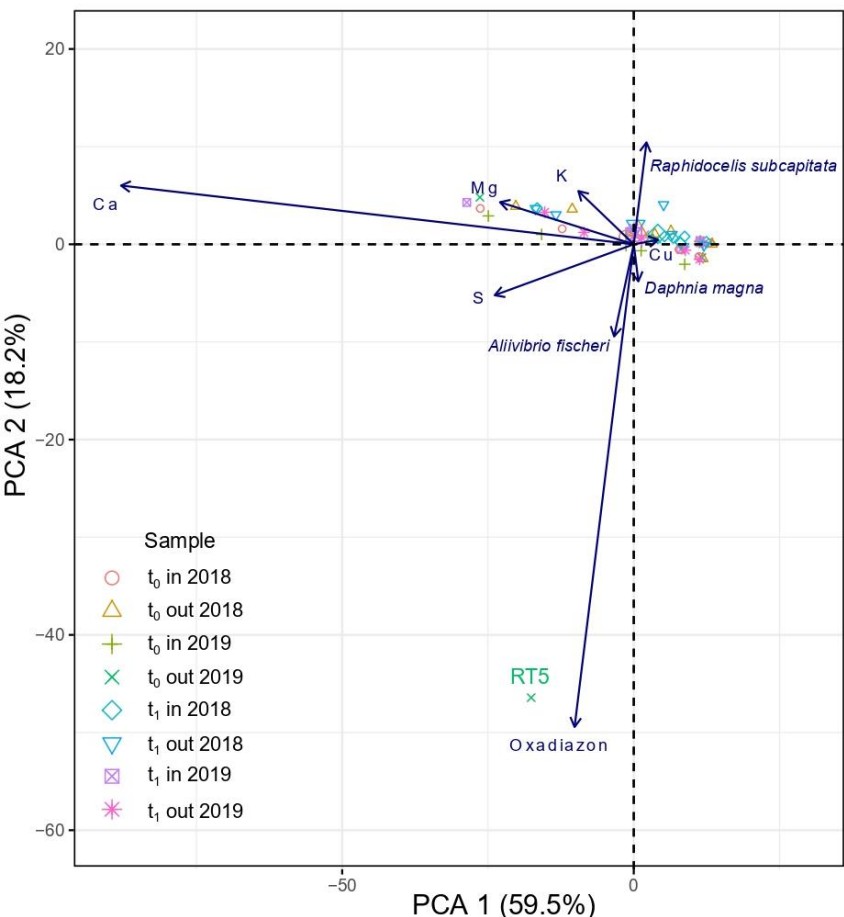

**Figure 2.** Plot of PCA applied to chemical and ecotoxicological characteristics of all water samples from rice paddies ($t_0$ and $t_1$ labels refer to the campaigns before and after PPPs treatment, respectively; in and out indicate the samples taken from streams entering and exiting the paddy water chamber, respectively, in 2018 and 2019. Only parameters (blue arrows) that contribute more than 15% in the variability of the samples are shown.

This herbicide is, therefore, the substance that most influences the characteristics of paddy water; its use is regularly reported by conventional paddies, while it has not been used by organic ones.

## 4. Conclusions

In conclusion, the study highlights the importance of assessing the impact of pesticides directly in paddy fields using an integrated approach, because the chemical or ecotoxicological approach alone is unable to consider numerous environmental factors that can occur as the photodecomposition responsible for rapid degradation [48], the volatilization favored

by high temperatures [49,50], the hydrolysis influenced by the flooded conditions and pH of the paddy water [51–53], and microbial degradation [54].

Chemical analyses of paddy waters revealed the presence of several PPPs, such as Oxadiazon and Metolachlor, in conventional fields. Relatively high levels of Oxadiazon were also detected in the biological fields, demonstrating its ubiquitous presence in the paddy rice area examined in this study. Its homogeneous concentration in the fields might be derived by the cascade circulation pattern of irrigation water in the area. Indeed, it is quite likely that the Oxadiazon applied to conventional rice paddies moves to organic from ones through the circulation of water, determining a relevant ecotoxicological response in both conventional and in organic fields. Nevertheless, it is reasonable to assume that the pesticide level decreases rapidly over time in the runoff water, accounting for volatilization, degradation, leaching to groundwater, and sorption to soil [49].

The application of a multidisciplinary approach has allowed for the identification of the main cause of contamination in rice fields in relation to ecotoxicological effects, i.e., Oxadiazon freely circulating in waters and the definition of a suitable solution strategy: the realization of organic districts with water circulation isolated from conventional fields should limit the transfer of the PPPs through the fields, enhancing the benefits of organic farming practices.

**Supplementary Materials:** The following supporting information can be downloaded at: https://www.mdpi.com/article/10.3390/w14244136/s1, Table S1: Weights assigned to the variables considered in the integrated index for estimating ecotoxicological Hazard Quotient ($HQ_{eco}$); Table S2: Chemical characterization of soil samples from paddies; Table S3: Concentration of PPPs measured in soil samples of paddies.

**Author Contributions:** Conceptualization, F.O., A.T., A.P. and C.M.; Formal analysis, F.O.; Funding acquisition, S.D.; Investigation, A.T., A.P., G.S., M.T.B., E.R. and C.M.; Methodology, F.O., A.T., A.P. and C.M.; Project administration, F.O., C.M., S.D. and E.R.; Resources, A.T., A.P., G.S., M.T.B. and C.M.; Supervision, F.O., C.M. and S.D.; Validation, F.O., A.T., A.P. and C.M.; Visualization, F.O.; Writing—original draft, F.O.; Writing—review and editing, A.T. and M.B. All authors have read and agreed to the published version of the manuscript.

**Funding:** This research was funded by the Italian Ministry of Ecological Transition from 2015 to 2020 as part of the project for the experimentation of the Measures of the National Action Plan (PAN) for the sustainable use of pesticides for the protection of biodiversity, especially in "Natura 2000" sites, in accordance to 2009/128/CE Directive, transposed in Italy by legislative decree n. 150/2012.

**Data Availability Statement:** Data supporting reported results can be found in: La sperimentazione dell'efficiacia delle Misure del Piano d'Azione nazionale per l'uso sostenibile dei prodotti fitosanitari (PAN) per la tutela della biodiversità, ISPRA, Vol. 330/2020, 392 pp. (https://www.isprambiente.gov.it/it/pubblicazioni/rapporti/rapporto-330-2020-web-12-01-2021-rid-1.pdf; accessed on 5 December 2022).

**Conflicts of Interest:** The authors declare no conflict of interest.

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
