# Peer review of "Ecotoxicological and Chemical Approach to Assessing Environmental Effects from Pesticide Use in Organic and Conventional Rice Paddies"

_water, doi:10.3390/w14244136_

Round 1

Reviewer 1 Report

The article "Ecotoxicological and chemical approach to assessing environmental effects from pesticide use in organic and conventional rice paddies" is well written. The methods are adequate. The results are discussed by the authors properly. I have no comments on the article and I think that it should be published in the journal "Water". 

1. What is the main question addressed by the research? Evaluate the environmental impact due to the use of PPPs by using an integrated approach based on chemical analyses and ecotoxicological hazard assessment, supported by statistical tools. 2. Do you consider the topic original or relevant in the field? Does it address a specific gap in the field? Yes, article is relevants. 3. What does it add to the subject area compared with other published material? New and relevant data on the research topic are presented. 4. What specific improvements should the authors consider regarding the methodology? What further controls should be considered? Improvements are not required. 5. Are the conclusions consistent with the evidence and arguments presented and do they address the main question posed? Yes. 6. Are the references appropriate? Yes. 7. Please include any additional comments on the tables and figures. Tables and figures are well done and do not require revision.

Author Response

Point 1: The article "Ecotoxicological and chemical approach to assessing environmental effects from pesticide use in organic and conventional rice paddies" is well written. The methods are adequate. The results are discussed by the authors properly. I have no comments on the article and I think that it should be published in the journal "Water".

  1. What is the main question addressed by the research? Evaluate the environmental impact due to the use of PPPs by using an integrated approach based on chemical analyses and ecotoxicological hazard assessment, supported by statistical tools. 2. Do you consider the topic original or relevant in the field? Does it address a specific gap in the field? Yes, article is relevants. 3. What does it add to the subject area compared with other published material? New and relevant data on the research topic are presented. 4. What specific improvements should the authors consider regarding the methodology? What further controls should be considered? Improvements are not required. 5. Are the conclusions consistent with the evidence and arguments presented and do they address the main question posed? Yes. 6. Are the references appropriate? Yes. 7. Please include any additional comments on the tables and figures. Tables and figures are well done and do not require revision.

Response 1: We are very grateful reviewer was very enthusiastic about our study.

Reviewer 2 Report

The manuscript is well organized to evaluate the environmental impact due to the use of PPPs by using an integrated approach based on chemical analyses and ecotoxicological hazard assessment. The following should be improved

1.     The large data expressed as Tables should be arranged as Supporting materials, to reduce the length of the manuscript.

2.     Figure 2 should be improved to make it more aesthetically, not an original draft.

3.     The Conclusions should focus on the main results of the research, but not a mini overview.

Author Response

The manuscript is well organized to evaluate the environmental impact due to the use of PPPs by using an integrated approach based on chemical analyses and ecotoxicological hazard assessment. The following should be improved

  1. The large data expressed as Tables should be arranged as Supporting materials, to reduce the length of the manuscript.

  1. Figure 2 should be improved to make it more aesthetically, not an original draft.

  1. The Conclusions should focus on the main results of the research, but not a mini overview.

Point 1: The large data expressed as Tables should be arranged as Supporting materials, to reduce the length of the manuscript.

Response 1: Table 3, Table 4 and Table 5 were eliminated in the main manuscript and inserted into the Supplemenatry materials document as Table SM 1, Table SM 2 and Table SM 3, respectively.

Point 2: Figure 2 should be improved to make it more aesthetically, not an original draft..

Response 2: Figure 2 was modified to render it more aesthetically appealing and effective

Figure 2 – Plot of PCA applied to chemical and ecotoxicological characteristics of all water samples from rice paddies (t0 and t1 labels refer to the campaigns before and after PPPs treatmen, respectively; in and out to samples taken from streams entering and exiting the paddy water chamber in 2018 and 2019. Only parameters (arrows) that contribute more than 15% on the variability of the samples are shown.

Point 3: The Conclusions should focus on the main results of the research, but not a mini overview.

Response 2: The conclusive section was revisioned as follows:

The conclusions have been reorganized as recommended, reducing the length of the paragraph and placing more emphasis on the main results obtained.

Reviewer 3 Report

The research design is complete and well described.

I only have one question: for the ecotoxicological assessment of soil, the biuluminescence inhibition assay on Aliivibrio fischeri was performed on soil eluate. Why the authors did not perform the same bioassay directly on soil (by using the solid phase test protocol)?

Author Response

Point 1: I only have one question: for the ecotoxicological assessment of soil, the biuluminescence inhibition assay on Aliivibrio fischeri was performed on soil eluate. Why the authors did not perform the same bioassay directly on soil (by using the solid phase test protocol)?

Response 1: The Solid Phase Test (SPT) protocol still presents problems with correct interpretation of the result, due to the loss of bacteria remaining attached to the solid particles of sample and osmotic environment not suitable to bacteria and solution instability problems due to the high pH values that often exceed the biological compatibility range.
These problems have currently been solved only for marine sediment samples.
So we preferred to adopt more standardized and proven methodologies.

We suggest consulting the following bibliography in this regard.

Bulich, A.A., Greene, M.W., Underwood, S.R., 1992. Measurement of soil and sediment toxicity to bioluminescent bacteria when in direct contact for a fixed time period. In: Water Environment federation 65th Annual Conference & Exposition, 20–24 September, New Orleans, Lousiana, USA.

 ONORATI F., Pellegrini D., Ausili A., 1998. Sediment toxicity assessment with Photobacterium phosphoreum: a preliminary evaluation of natural matrix effect. Fresenius Environ. Bull., 7: 596-60.

ONORATI F., Volpi Ghirardini A., 2001. Informazioni fornite dalle diverse matrici da testare con i saggi biologici: applicabilità di Vibrio fischeri. Biol. Mar. Medit., 8(2): 31-40.

ONORATI F., Mecozzi M., 2004. Effects of two diluents in the Microtox® toxicity bioassay with marine sediments. Chemosphere, 54: 679 – 687.

Martini P., ONORATI F., Ferrari C.R., Ardesi E., 2009. Valutazione della tossicità naturale con saggio Microtox® in fase solida per campioni sabbiosi: proposta di una specifica curva di normalizzazione pelitica. Biol. Mar. Mediterr., 14(2): 103-109.
